# Influence of Season, Storage Temperature and Time of Sample Collection in Pancreatitis-Associated Protein-Based Algorithms for Newborn Screening for Cystic Fibrosis

**DOI:** 10.3390/ijns10010005

**Published:** 2024-01-12

**Authors:** Pia Maier, Sumathy Jeyaweerasinkam, Janina Eberhard, Lina Soueidan, Susanne Hämmerling, Dirk Kohlmüller, Patrik Feyh, Gwendolyn Gramer, Sven F. Garbade, Georg F. Hoffmann, Jürgen G. Okun, Olaf Sommerburg

**Affiliations:** 1Centre for Paediatric and Adolescent Medicine, Department of General Paediatrics, Division of Neuropaediatrics and Metabolic Medicine, University Hospital Heidelberg, Im Neuenheimer Feld 430, 69120 Heidelberg, Germanyg.gramer@uke.de (G.G.); georg.hoffmann@med.uni-heidelberg.de (G.F.H.); 2Department of Paediatrics, SLK-Kliniken Heilbronn GmbH, Am Gesundbrunnen 20–26, 74078 Heilbronn, Germany; sumathy-jeya@web.de; 3Centre for Paediatric and Adolescent Medicine, Department of Neonatology, University Hospital Heidelberg, Im Neuenheimer Feld 430, 69120 Heidelberg, Germany; janina.eberhard@med.uni-heidelberg.de; 4Dietmar-Hopp Centre for Metabolic Diseases Heidelberg, University Hospital Heidelberg, Im Neuenheimer Feld 669, 69120 Heidelberg, Germany; lina.soueidan@med.uni-heidelberg.de (L.S.); dirk.kohlmueller@med.uni-heidelberg.de (D.K.); patrik.feyh@med.uni-heidelberg.de (P.F.); sven.garbade@med.uni-heidelberg.de (S.F.G.); juergenguenther.okun@med.uni-heidelberg.de (J.G.O.); 5Centre for Paediatric and Adolescent Medicine, Division of Pediatric Pulmonology & Allergy and Cystic Fibrosis Center, University Hospital Heidelberg, Im Neuenheimer Feld 430, 69120 Heidelberg, Germany; susanne.haemmerling@med.uni-heidelberg.de; 6Translational Lung Research Center (TLRC), German Lung Research Center (DZL), University of Heidelberg, Im Neuenheimer Feld 130.3, 69120 Heidelberg, Germany; 7University Children’s Hospital, University Medical Center Hamburg-Eppendorf, Martinistraße 52, 20246 Hamburg, Germany

**Keywords:** cystic fibrosis, newborn screening, PAP, seasonal effect, temperature

## Abstract

Newborn screening (NBS) for cystic fibrosis (CF) based on pancreatitis-associated protein (PAP) has been performed for several years. While some influencing factors are known, there is currently a lack of information on the influence of seasonal temperature on PAP determination or on the course of PAP blood concentration in infants during the first year of life. Using data from two PAP studies at the Heidelberg NBS centre and storage experiments, we compared PAP determinations in summer and winter and determined the direct influence of temperature. In addition, PAP concentrations measured in CF-NBS, between days 21–35 and 36–365, were compared. Over a 7-year period, we found no significant differences between PAP concentrations determined in summer or winter. We also found no differences in PAP determination after 8 days of storage at 4 °C, room temperature or 37 °C. When stored for up to 3 months, PAP samples remained stable at 4 °C, but not at room temperature (*p* = 0.007). After birth, PAP in neonatal blood showed a significant increasing trend up to the 96th hour of life (*p* < 0.0001). During the first year of life, blood PAP concentrations continued to increase in both CF- (36–72 h vs. 36–365 d *p* < 0.0001) and non-CF infants (36–72 h vs. 36–365 d *p* < 0.0001). Seasonal effects in central Europe appear to have a limited impact on PAP determination. The impact of the increase in blood PAP during the critical period for CF-NBS and beyond on the applicability and performance of PAP-based CF-NBS algorithms needs to be re-discussed.

## 1. Introduction

Following agreement on the benefits of newborn screening (NBS) for cystic fibrosis (CF) (e.g., [1]), nationwide CF-NBS was introduced in Germany on 1 September 2016 after a long decision-making process. One of the problems discussed was the uncertainty as to whether and how, in accordance with the German Genetic Diagnostics Act [2], a CF-NBS algorithm searching for CFTR variants could be used in Germany. In this context, a purely biochemical CF-NBS algorithm published by Sarles et al. in 2005 [3,4], using immunoreactive trypsinogen (IRT) as the first tier and pancreatitis-associated protein (PAP) as the second tier, was considered as an alternative to genetic CF-NBS. In order to evaluate the feasibility of such a CF-NBS algorithm under the conditions in Germany, one of two German pilot studies was conducted with a modified IRT/PAP protocol at the CF-NBS centre in Heidelberg (screening of 110,000 newborns per year from southwest Germany) [5]. During this pilot study, which started in 2008 and was conducted until the start of the Germany-wide CF-NBS programme on 1 September 2016, a total of 480,151 newborns were screened for CF. This showed that purely biochemical, PAP-based CF-NBS algorithms achieve an acceptable sensitivity, but only a very low positive predictive value (PPV). Therefore, in the current German CF-NBS programme, a three-stage CF-NBS algorithm is used, in which a search for 31 *CFTR* variants is included as a third tier. Apart from Germany, PAP-based CF-NBS algorithms are also used in the Netherlands [6,7], Portugal [8] and Austria [9], among others, because IRT/PAP algorithms seem to have advantages over genetic algorithms in screening multi-ethnic populations and, depending on the design of the algorithm, the number of false detections of healthy carriers or newborns designated as CFSPID after CF-NBS can be significantly reduced.

When PAP-based CF-NBS was introduced in the Netherlands, valuable data had already been collected on influencing factors such as sex, gestational age, birth weight, blood transfusion and timing of heel prick [10]. There are also some data on PAP concentrations in CF patients at later time points after CF-NBS, e.g., from a PAP pilot study carried out in East Saxony, Germany [11]. It should be noted, however, that although PAP has been used in CF-NBS for more than 10 years, there are still discussions about the sensitivity of this parameter, which may be related to the fact that there are not yet sufficient, generally valid, reliable data in the literature on influencing factors such as climatic fluctuations in the use of IRT/PAP or the course of PAP concentration as such in the blood of infants during the first days of life and the entire first year of life. However, considering that in Germany, CF-NBS can be performed up to the 28th day if the parents of an infant refuse it as part of the routine NBS in the first days of life [2] or that in the Netherlands, samples for CF-NBS sent up to the end of the sixth month of life are accepted, it would be important to know the performance of this method in the later period. In order to answer this question, the Heidelberg Late-IRT&PAP Study was established, in which IRT and PAP were measured again at later time points in healthy newborns, in newborns with a positive CF-NBS and in infants diagnosed with CF. If available, these values were then compared with those at the time of the regular CF-NBS (36–72 h of life), which provided an opportunity to obtain longitudinal trajectories of PAP concentrations from the early time of the first CF-NBS, in the period from 20 to 35 days of life and beyond until the end of the first year of life. Another part of this work was to try to evaluate the temperature dependence of the IRT/PAP protocol from the available data of the Heidelberg NBS centre in order to assess the applicability of IRT/PAP protocols under real-life conditions. Since the conditions in NBS laboratories can be designed to be largely independent of seasonal influences, the main objective was to investigate the influence of temperature from sample collection to arrival at the NBS laboratory using storage experiments.

## 2. Materials and Methods

### 2.1. Study Population, Test Methods and CF-NBS Algorithms

Values of IRT and PAP determinations were collected from April 2008 to January 2023 at the Heidelberg NBS centre in two different CF-NBS programmes, each with its own PAP-based algorithm. From 2008 to September 2016, the Heidelberg CF-NBS pilot study was conducted using a two-stage IRT/PAP algorithm with an IRT-dependent safety net (SN) (IRT/PAP + SN, Figure A1a in Appendix A). Since 2016, the routine CF-NBS programme in Germany has been conducted with a three-stage IRT/PAP/DNA algorithm, which is also conducted with an IRT-dependent SN (IRT/PAP + SN/DNA, Figure A1b in Appendix A). Dried blood spots (DBSs) are sampled for NBS in Germany between 36 and 72 h after birth and reach the NBS centre by courier or regular postal service within 24–48 h. From 2018 to 2023, the Late-IRT&PAP study was conducted in parallel to the current routine CF-NBS. All newborns diagnosed as CF patients after CF-NBS and whose clinical follow-up data were used for the present work were previously included in a longitudinal register study (Track-CF). All studies described here were approved by the Ethics Committee of the Medical Faculty of Heidelberg University: CF-NBS pilot study S 337/2007, Late-IRT&PAP S-268/2017, Track-CF S-211/2011.

IRT testing was carried out according to Bowling and Bowling [12]. From 2008 until 2015, IRT was detected using the AutoDELFIA^®^Neonatal IRT kit (Perkin-Elmer, Turku, Finland) according to the manufacturer’s protocol. From 2015, IRT measurement was changed to the Genetic Screening Processor Workstation (GSP^®^)(Perkin-Elmer, Turku, Finland), which also required the IRT cut-off to be adjusted. IRT measurement was used as the first tier using a floating IRT cut-off (≥99.0th percentile) for initiation of second-tier testing. An ultrahigh IRT value, which was used as a SN, was defined as ≥99.9th percentile.

PAP testing was always assayed in duplicate by an enzyme-linked immunosorbent assay (ELISA). From 2008 until September 2016, the MucoPAP kit (DYNABIO S.A., Marseille, France) was used, and from October 2016 until now, the MucoPAP-F kit (DYNABIO S.A., Marseille, France) has been used, following the manufacturer’s instructions [4,13]. For CF-NBS, PAP testing relies on one PAP cut-off using the lower PAP cut-off value of the two IRT-dependent PAP cut-off values originally published by Sarles et al. [3], which were defined at ≥1.6 μg/L when using MucoPAP and ≥2.2 μg/L after switching to MucoPAP-F. From 2008, a SN strategy was applied, which comes into operation when the initial IRT is measured as ultra-high (≥99.9th percentile), regardless of the measured PAP value [5]. For better comparability for this publication, we have normalised all PAP values from the older specifications to the limit value for MucoPAP-F of 2.2 μg/L that now applies in the Heidelberg NBS centre.

In both CF-NBS algorithms, determination of IRT and PAP was performed consecutively according to an algorithm by Sommerburg O. et al. [5] that was modified compared to the first description by Sarles J. et al. [3]. From 2008 until 2016, the IRT/PAP + SN algorithm was performed as a pure biochemical protocol. According to that, CF-NBS was considered positive if either PAP testing was positive (≥1.6 μg/L) or an ultra-high IRT (≥99.9th percentile) was present (Figure A1a). The three-stage IRT/PAP + SN/DNA algorithm introduced in Germany in 2016 has continued to use the modifications of the PAP determination from the Heidelberg pilot study [5,14,15], but if PAP is positive, a search for 31 *CFTR* variants is performed in the third algorithm tier [2]. However, in this CF-NBS algorithm, PAP is only determined in neonates whose initial IRT is between the 90.0th and 99.9th percentiles. If the IRT is ≥99.9th percentile, the conditions for the SN are fulfilled and these newborns are immediately evaluated as CF-NBS positive (Figure A1b).

### 2.2. Study Design

#### 2.2.1. Evaluation of a Possible Seasonal Effect on IRT/PAP Determination

For a generally valid answer to the question of seasonal effects in the determination of blood PAP concentrations in the CF-NBS, the data from the Heidelberg pilot study (2008–2016) were used, because the PAP values collected in the current German CF-NBS algorithm might be subject to a selection bias, in our opinion, due to the limited PAP determination only in newborns with IRT between the 99.0th and 99.9th percentiles. Therefore, the same data set was also used to evaluate seasonal effects on IRT to be determined beforehand. However, as the Heidelberg NBS laboratory moved twice from 2008 to 2016, only the years 2009 to 2015 were considered and used to assess the seasonal effect on PAP determination. In addition, the IRT measurement was changed from the Autodelfia^®^ method to the GSP^®^ method in 2014, so only the years 2009 to 2013 were considered here to ensure equal conditions. In order to compare the extremes of the cold and warm seasons, the summer months June, July and August were compared with the winter months December, January and February. Thus, the term “winter of year X” means December of year X and the months January and February of the following year, i.e., year X + 1. For IRT, the mean values and the 95.0th, 99.0th and 99.9th percentiles for the respective months were calculated and compared with each other. For PAP, the mean values and standard deviations determined were calculated and then compared with each other. The statistical evaluation was carried out using the two-factor ANOVA test with the factors “year” and “season”.

#### 2.2.2. Storage Tests on Possible Temperature Effects on PAP Determination

In many NBS centres, PAP is measured only once a week. To simulate the effect of the average time between blood collection and PAP measurement in the screening laboratory, a storage period of 8 days was chosen. Anonymised punchings from screening cards of the ongoing 2017 CF-NBS were used for the study. For an initial storage experiment with different temperatures, 5 punchings from screening cards each with a PAP value above the PAP cut-off (>2.2 μg/L) and 5 punchings each with a PAP value below the PAP cut-off (<2.2 μg/L) were randomly selected for storage at 4 °C, room temperature (RT, 20 °C) and 37 °C. As it is also mandatory in Germany to keep screening cards from the NBS for three months in order to be able to carry out any necessary repeat measurements, another set of these punchings was also stored at 4 °C for three months. Statistical analysis was performed with a linear mixed model with the criterion “PAP value” and the variable “condition”. In addition, the longer-term influence of RT was investigated in another experiment. For this purpose, one anonymised sample with a PAP value below the PAP cut-off value after the initial measurement of DBS was obtained daily for 30 days and stored at RT in a dark environment. In addition, punchings of 6 screening cards with PAP values above the PAP cut-off value were randomly collected and stored under the same conditions, resulting in a total of n = 36 samples. Three months after the start of the storage trial at RT, PAP was measured again from the punchings. At this time, each punching had been at RT for between 2 and 3 months. Statistical analysis was performed using a paired *t*-test.

#### 2.2.3. Course of Blood PAP Concentrations in the First Days of Life

For a universal answer to the question about the time course of blood PAP concentrations in infants in the first days of life, the data set of the Heidelberg pilot study (2008–2016) was used again, due to the selection bias in the current German CF-NBS described above. The available data from the pilot study including data from late and early sampling were summarised in 12 h intervals and presented in box–whisker diagrams. Quantile regression and percentile bootstrap methods were used to compare arbitrary quantiles between groups, while needed *p*-values were adjusted for type 1 error rate with Hochberg correction. An Aligned Rank Transformation (ART) non-parametric ANOVA [16] was used to test for changes between the different heel prick times.

#### 2.2.4. Course of Blood PAP Concentrations from 20th to the 365th Day of Life

The Late-IRT&PAP study was conducted in parallel to routine CF-NBS at the NBS centre in Heidelberg with the support of some university CF centres in southwest Germany. First, DBSs were collected for the determination of IRT and PAP from outpatients and inpatients from the age of 21 up to 365 days of age. Only healthy subjects or those with mild disease were included who had a corrected age of 36 + 0 weeks’ gestation at the time of the study’s blood draw and who had no medical disease that could potentially affect PAP and IRT values. To assess the appropriateness of a late PAP determination for the purpose of CF-NBS, this group was further differentiated into infants between 21 and 35 days of life and infants from 36 to 365 days of life. Secondly, CF-NBS-positive infants who presented to CF centres for diagnostic confirmation (sweat chloride determination and clinical assessment) after a positive CF-NBS had DBS sampling again for IRT and PAP determination. If a CF diagnosis was confirmed, these newborns were assigned to the “CF” group; if CF was excluded, these newborns were assigned to the “Non-CF” group together with the group of infants recruited at the clinic with no known or suspected CF (see also Figure A2 in Appendix A). From all infants participating in the Late-IRT&PAP study, an attempt was made to obtain the blood PAP value from the CF-NBS in order to present a longitudinal course of the individual PAP concentration. For statistical analysis of the PAP groups, the Mann–Whitney test was applied.

## 3. Results

### 3.1. Evaluation of a Possible Seasonal Effect on IRT/PAP Determination

Out of the data set from the Heidelberg CF-NBS pilot study, IRT data from the years 2009 to 2013 (only data detected with Auto-Delfia^®^) and PAP data from 2009 to 2015 were used for evaluation. The IRT values generally showed slightly higher averaged concentrations in the winter months than in the summer months (Table A1 in Appendix A). In 4 out of the 5 years, the differences were statistically significant (*p* < 0.001 in 2009, 2010, 2012; *p* < 0.01 in 2013). The results for the more relevant 95.0th, 99.0th and 99.9th percentiles are given in Table 1. For the 99.0th percentile, which is critical for the CF-NBS algorithm in Germany, the IRT values tended also to be higher in winter than in summer, but only the values from 2009 to 2011 were statistically significant (Table 1). For the 99.9th percentile, there were no significant differences (Table 1).

The mean PAP values in the years 2010 to 2015 were also higher in winter than in summer. Only in 2009 was the mean PAP value lower in winter than in summer. However, no measured differences were statistically significant (Figure 1, Table A2).

### 3.2. Direct Temperature Effect on PAP Determination

The samples from the five screening cards whose initial PAP measurement was above the PAP cut-off value and from those whose PAP value was below the PAP cut-off value showed no significant change in concentration for PAP after eight days of storage at 4 °C (*p* = 0.3947), RT (*p* = 0.8802), or 37 °C (*p* = 0.2961). Even after three months of storage at 4 °C, no significant decrease in PAP concentration could be detected compared to the initial PAP measurement (*p* = 0.1801). The data are shown in Figure 2 and Table A3 in Appendix A. However, in the second storage experiment, where the punchings were stored at RT for 2 to 3 months, the samples showed a significant decrease in PAP concentration compared to the initial measurement (*p* = 0.007), as shown in Table A3 in Appendix A.

### 3.3. Course of Blood PAP Concentrations in the First Days of Life

From the Heidelberg pilot study (2008–2016), the mean PAP values of 3421 newborns were available for this analysis. In 2909 newborns, blood samples were taken between 36 and 72 h of life, as required by the German NBS. For medical reasons, blood samples were taken early in 191 newborns and late in 321 newborns. For our study, we considered the PAP values divided into 12 h intervals up to the 96th hour of life and thus excluded the PAP values of 85 newborns whose blood samples were taken after the 96th hour of life. Immediately after birth, PAP values were lowest and then showed an increasing trend (Figure 3). The 0–12 h cluster differed significantly from the 36–48 h (*p* = 0.0092), 48–60 h (*p* < 0.0001), 60–72 h (*p* < 0.0001), 72–84 h (*p* = 0.0005) and 84–96 h (*p* = 0.0003) clusters. Also, in the blood sampling interval between 36 and 72 h of life, which is crucial for NBS in Germany, mean PAP levels increased from 0.75 μg/L in the 36–48 h interval to 1.00 μg/L in the 60–72 h interval (Figure 3). PAP values in the 36–48 h interval were significantly different from those in the 48–60 h interval (*p* < 0.0001) and those in the 60–72 h interval (*p* < 0.0001).

### 3.4. Course of Blood PAP Concentrations in the First Year of Life

For this question, the normalised PAP values from the “36 to 72 h of life” time window intended for the CF-NBS from the Heidelberg pilot study were compared with the more recent data from the Late-IRT&PAP study for the “21 to 35 days of life” and “36 to 365 days of life” time windows. During the Late-IRT&PAP-study, we recruited 88 infants with no known or suspected CF and 89 infants with positive CF-NBS who were seen outpatient for diagnostic confirmation. Furthermore, we included 16 already known CF patients under 365 days of age. In total, nine infants had to be excluded because of insufficient material for analysis, eight because they turned out to be outside the age range defined by the inclusion criteria of the study, and eight because insufficient information was provided with the sample for analysis. Three patients in the non-CF group were excluded because they developed severe disease (two end-stage renal disease, and one metabolic disease), as an influence of the disease on the PAP value could not be excluded with certainty. The final cohort of infants with positive CF-NBS consisted of 77 infants, which were grouped, as described before, as “Non-CF” infants or “CF” infants. In the end, two groups were formed for further analysis: “Non-CF” with 124 children and “CF” with 38 children. A more detailed overview also about the distribution of the different time intervals is given in Figure A2 in Appendix A.

The results show that PAP concentrations in the blood of infants increase significantly during the first year of life. Compared to the time of CF-NBS, PAP levels in both non-CF infants and infants with CF were significantly higher between 21 and 35 days of life (*p* < 0.0001, *p* = 0.0001, respectively) and between 36 and 365 days of life (*p* < 0.0001, *p* < 0.0001, respectively). In addition, the vast majority of infants aged between 21 and 365 days had a PAP blood concentration significantly higher than the PAP cut-off value of 2.2 µg/L set for CF-NBS in Germany. In non-CF infants, PAP values measured between 36 and 365 days of age were again significantly higher than those measured between 21 and 35 days of age (*p* = 0.0006) (Figure 4A,B; Table A4 in Appendix A).

For seventeen of the non-CF newborns and for seven of the infants with CF, PAP values from the CF-NBS performed between 36 and 72 h of life could be assigned to the PAP values from the Late-IRT&PAP study. Figure 5 shows these individual curves for infants with CF and healthy non-CF infants. Again, PAP levels increase in CF patients shortly after birth and during the first year of life. However, this general trend is also observed in non-CF infants, although five out of seventeen non-CF infants in our study showed a decrease in PAP levels at 21–35 days compared to CF-NBS at 36–72 h.

## 4. Discussion

PAP-based CF-NBS algorithms have been used in various screening programmes for more than a decade. To the best of our knowledge, our work was the first to systematically address the issues of “seasonal” and “temperature” effects as well as “time of sampling” in the context of PAP-based CF-NBS.

### 4.1. Evaluation of Climate and Temperature Effects on the IRT/PAP Determination

Although our investigations were primarily aimed at generating missing knowledge for the determination of PAP, the climatic influence on both IRT and PAP must be considered with regard to the combined use of both parameters in CF-NBS algorithms. For IRT, deviations in the mean values of all IRT measurements between winter and summer have previously been shown for the North American continental climate in a 10-year cycle [17]. Our 5-year data generally support these observations, although the changes in terms of slightly higher mean values in winter compared to summer were statistically significant only in 4 out of 5 years (Table A1 in Appendix A), and for the decision-critical percentiles 95.0 and 99.0, only in 3 out of 5 years (Table 1). There were no significant differences at the 99.9th percentile (Table 1). At present, we believe that when using a floating cut-off value for the IRT, we can assume that the small differences in IRT determination between winter and summer in countries where the CF-NBS is conducted have no practical significance. However, if a “fixed” IRT cut-off value is used and set too narrowly, the differences observed could certainly lead to false-negative assessments of screening cases. One way to avoid this problem would be to have a sufficient “safety margin” when determining the IRT cut-off in the first step of the CF-NBS algorithm or to use IRT values collected during the summer months as a benchmark for determining the IRT cut-off.

To our knowledge, such data are not yet available for PAP. Our data, obtained over 7 years (2009–2015), show that the averaged PAP values also tended to be slightly lower in summer than in winter in 6 out of 7 years (Figure 1, Table A2 in Appendix A), but the differences did not reach statistical significance. We therefore assume that one or two fixed PAP cut-off values, as currently used in various CF-NBS programmes, can be considered uncritical with regard to seasonal influences under Central European conditions (e.g., Germany, Netherlands, Austria). Our experiments on the effect of temperature on PAP determination under controlled conditions also support this observation. Here, the influence of temperature was simulated over 8 days to cover the time frame from sampling to determination in the NBS laboratory. The experiment showed that the measured PAP values for both punchings with PAP values above the PAP cut-off value and punchings with PAP values below the PAP cut-off value in the initial measurement showed no significant decrease in concentration after storage at 4 °C, RT (20 °C) or even 37 °C. This is particularly noteworthy at 20 °C and 37 °C as it supports the previous practice of transporting samples without refrigeration. However, we would like to point out that with increasing climate change, these differences may become more pronounced and consequences for the storage and transport of screening cards will have to be drawn. In the tests where the samples were stored longer, however, differences occurred. While the PAP values obtained from the samples stored for three months at 4 °C showed no significant decrease in concentration, the samples stored for 2–3 months at RT showed a significant (*p* < 0.01) decrease in the concentration of PAP (Table A3 in Appendix A), which can be explained by denaturation processes during prolonged exposure to warm temperatures. The results thus support the practice required in Germany of storing newborn screening cards at 4 °C for three months for follow-up measurements. Storage of screening cards at RT for this period should not occur in view of the PAP parameter.

### 4.2. Influence of Time of Sampling on PAP Value

CF-NBS using PAP has been investigated before in several pilot studies. However, some studies concluded that the sensitivity of PAP was not high enough, either because a higher performance could be achieved when compared to a genetically based algorithm (e.g., in the Czech Republic [18]), or because it was found that PAP only increased to a range suitable for CF-NBS from the third day of life (Ranieri E., personal communication). Using data from the Heidelberg Pilot Study (2008–2016) and the Late-IRT&PAP Study (2017–2023), we examined the course of PAP concentrations in the blood of newborns from the first hours of life to the end of the first year of life. For PAP levels from 0 to 96 h of age, a sufficient data set was available from the Heidelberg pilot study. To investigate the influence of the time of blood collection, we divided the data into 12 h intervals, all of which were found to be statistically significantly different from each other. While PAP concentrations were low when blood was collected immediately after birth, the 12 h intervals up to 72 h of age showed a clear upward trend (Figure 3). Interestingly, this trend of increasing PAP concentrations was also significant in the period currently recommended for NBS in Germany (36 to 72 h of life). In contrast, the PAP values measured after 72 h of life no longer showed a consistent trend, which could be due to the lower number of samples and/or the underlying clinical reasons for the late sampling of the neonates. These data confirm the reported observations of an increase during this critical period for CF-NBS from a long-ago Australian pilot study (Ranieri E., personal communication) but also from results reported on a larger scale when the Dutch PAP-based CF-NBS was developed [10]. Whether these differences are also relevant for CF-NBS within 36–72 h should be discussed in future studies with data from CF patients with false-negative results from active CF-NBS programmes with PAP-based algorithms. However, the recently published final data from the Heidelberg pilot study [19] do not yet support this assumption. The low PAP values measured between the 24th and 36th hour of life, albeit with only a small number of values (n = 72), can neither invalidate nor confirm the assumption that the sensitivity of PAP-based CF-NBS algorithms is no longer sufficient at blood sampling times earlier than 36 h (Figure 3). However, according to the trend seen, it is questionable whether a PAP-based CF-NBS algorithm can be safely used earlier than 36 h with the fixed cut-off values applied so far. This would be the case, for example, in CF-NBS programmes in the USA, where blood for NBS is drawn from the 24th hour of life (e.g., [20]). However, this should also be considered for NBS programmes where, as in Germany, there is a discussion about bringing the time of sampling forward due to the advantages of NBS for metabolic diseases. Theoretically, this problem could again be solved by lowering the PAP threshold as shown before [13] or even using time-dependent PAP thresholds, but this would again require a third protocol tier (e.g., DNA) to ensure sufficient performance of the algorithm. However, knowing the trend of increasing PAP concentrations in the first days of life, the question of the relatively low PPV of purely biochemical IRT/PAP algorithms needs to be re-discussed, as it does not seem implausible that setting the PAP cut-off for the early samples between 36 and 72 h would require such a low PAP cut-off, which would then explain a high number of false-positive cases in the later samples [11,13,18,19].

As it is possible in Germany to perform CF-NBS up to the 28th day of life [2], it would be necessary, analogous to IRT, to thoroughly investigate the further course of the PAP concentration in the blood of infants until the 28th day of life. The results of our Late-IRT&PAP study provide insight into a later screening window from 21 to 35 days of age. However, our results show that the PAP values in the period between 21 and 35 days of age are significantly higher than in the period between 36 and 72 h of age (Figure 4 and Figure 5). Although it can be stated that all CF patients would have been detected in this period with the PAP cut-off value currently used in Germany, 68.9% of all non-CF infants would have had a false-positive CF-NBS result. This shows that PAP measurement with the currently used cut-off value is not a suitable method for CF-NBS in this period. However, it remains questionable whether it is useful to evaluate a cut-off value for PAP for this time window, as the single IRT measurement in this period is already a suitable parameter for which there is sufficient experience. On the other hand, it is questionable whether it is even necessary in Germany to perform the CF-NBS until the 28th day of life. This is because the indication would only be given if the parents had not expressly consented to CF-NBS during the routine NBS, as it is unfortunately not possible in Germany to give this consent retrospectively. If this option were available, one could simply use the screening card from the NBS, which is stored for 3 months anyway. Finally, our data showed that storage at 4 °C for 3 months would not adversely affect the results of IRT/PAP measurement. Therefore, it would be methodologically justifiable to perform a follow-up measurement of IRT and PAP on this sample up to 28 days after the first NBS, if the parents decide to do so after initially refusing the CF-NBS.

### 4.3. Limitations

There are some limitations to our work. First, it is important to note that the investigations of climatic influences on IRT and PAP were carried out with data collected several years ago. However, we still consider the use of these data to be appropriate because the basic conditions regarding collection, storage and transport of the screening card as well as the requirements determined by other target diseases have not changed in recent years. Secondly, the comparisons of PAP values at different sampling times had to be based on values from two different screening programmes performed with different PAP kits. Nevertheless, we consider this comparison to be meaningful because, before switching from the MucoPAP to the MucoPAP-F kit, comparative measurements were performed in our Heidelberg NBS centre and in two other German NBS centres, from which a valid conversion factor could be calculated. Furthermore, a similar comparison has already been scientifically accepted in another peer-reviewed publication [11]. Thirdly, we had actually planned a higher recruitment number for the Late-IRT&PAP study, especially in view of the relatively small number of neonates in whom longitudinal comparisons of PAP values from the Late-IRT&PAP study with those from routine CF-NBS were possible. However, it should be noted that in the German CF-NBS, such comparisons are only possible for 0.9% of all infants, namely those whose IRT value is above the 99.0th percentile but below the 99.9th percentile [2]. For all others, no PAP value is obtained. In addition, recruitment during the Late-IRT&PAP study was also severely hampered in the meantime by the COVID-19 pandemic.

## Figures and Tables

**Figure 1 IJNS-10-00005-f001:**
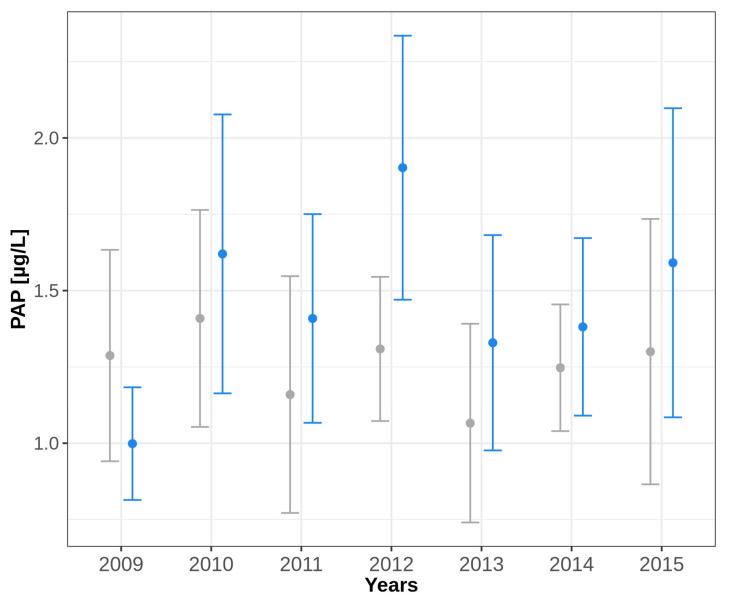
Seasonal effect on PAP determination in the CF-NBS. Values from 2009 to 2015 from the Heidelberg NBS centre (summer grey, winter blue) are presented as means with 95% confidence intervals. Summer includes the months June, July and August of the respective year “X”, winter includes the months December of the year “X” and January and February of the year “X + 1”.

**Figure 2 IJNS-10-00005-f002:**
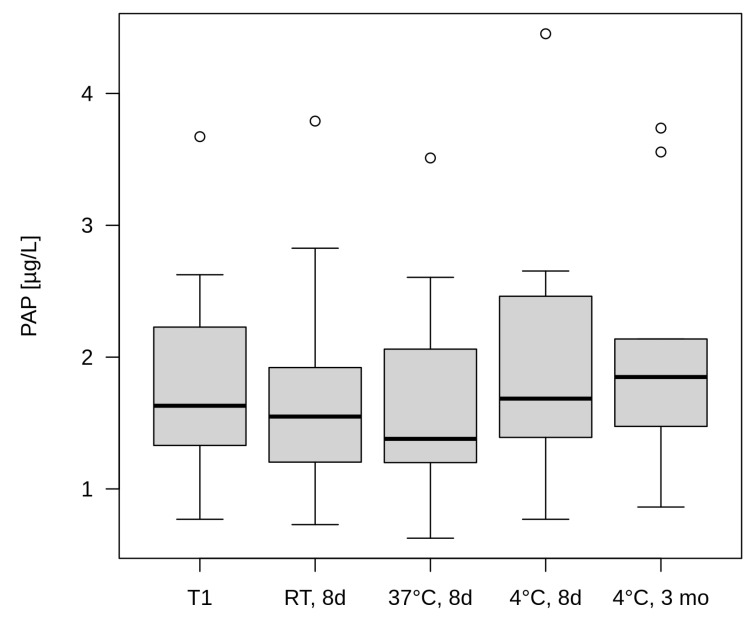
Change in PAP concentration in punchings of 10 screening cards (5 with PAP > PAP cut-off value and 5 with PAP < PAP cut-off value) at different storage temperatures shown as box plots with 25th to 75th percentiles. Whiskers correspond to 1.5-fold deviation from 25th or 75th percentile. The individual values above (circles) are outliers outside 1.5-fold deviation. T1: initial measurement, RT, 8 d: storage at room temperature for 8 days, 37 °C, 8 d: storage at 37 °C for 8 days, 4 °C, 8 d: storage at 4 °C for 8 days, 4 °C, 3 mo: storage at 4 °C for 3 months.

**Figure 3 IJNS-10-00005-f003:**
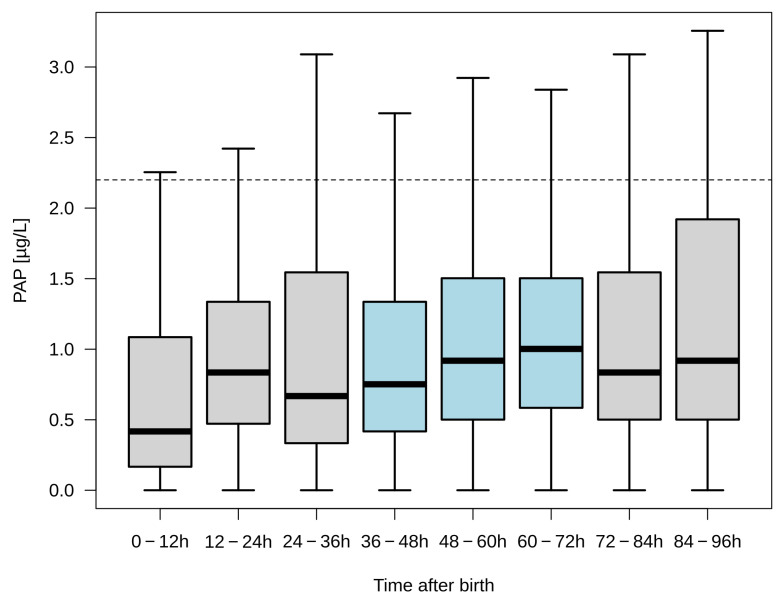
PAP concentrations in dried blood collected between 0 and 96 h after birth. PAP values (μg/L) are summarised in 12 h intervals and displayed as box plots with 25th to 75th percentiles. The dashed line represents the current PAP cut-off value (2.2 µg/L). The light blue boxes describe the period in which dry blood is sampled for NBS in Germany. Number of samples per interval: 0–12 h *n* = 92, 12–24 h *n* = 27, 24–36 h *n* = 72, 36–48 h *n* = 1155, 48–60 h *n* = 1123, 60–72 h *n* = 631, 72–84 h *n* = 183, 84–96 h *n* = 53. Significances: 0–12 h vs. 36–48 h *p* < 0.01, vs. 48–60 h *p* < 0.0001, vs. 60–72 h *p* < 0.0001, vs. 72–84 h *p* < 0.001, and vs. 84–96 h *p* < 0.001; 36–48 h vs. 48–60 h *p* < 0.0001 and vs. 60–72 h *p* < 0.0001.

**Figure 4 IJNS-10-00005-f004:**
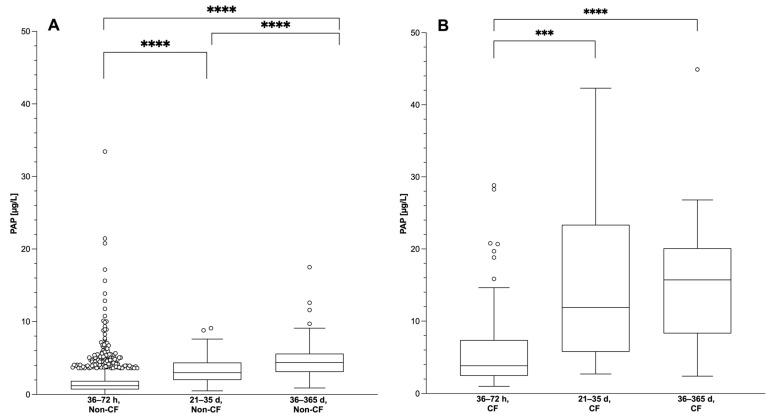
PAP values (μg/L) of (**A**) healthy infants and (**B**) infants with CF during different periods (36–72 h, 21–35 days, 36–365 days of life) shown as box plots with 25th to 75th percentiles. Whiskers correspond to 1.5-fold deviation from 25th or 75th percentile. The individual values above or below are outliers outside 1.5-fold deviation. Note: PAP values for the period 36–72 h of life are from the Heidelberg PAP pilot study (2008–2016). The PAP values were normalised to be comparable with the values from the other two periods obtained during the Late-IRT&PAP study (2018–2023). Non-CF: 36–72 h: *n* = 2301, 21–35 d: *n* = 61, 36–365 d: *n* = 63. CF: 36–72 h: *n* = 92, 21–35 d: *n* = 17, 36–365 d: *n* = 21. (**** *p* < 0.0001, *** *p* < 0.001).

**Figure 5 IJNS-10-00005-f005:**
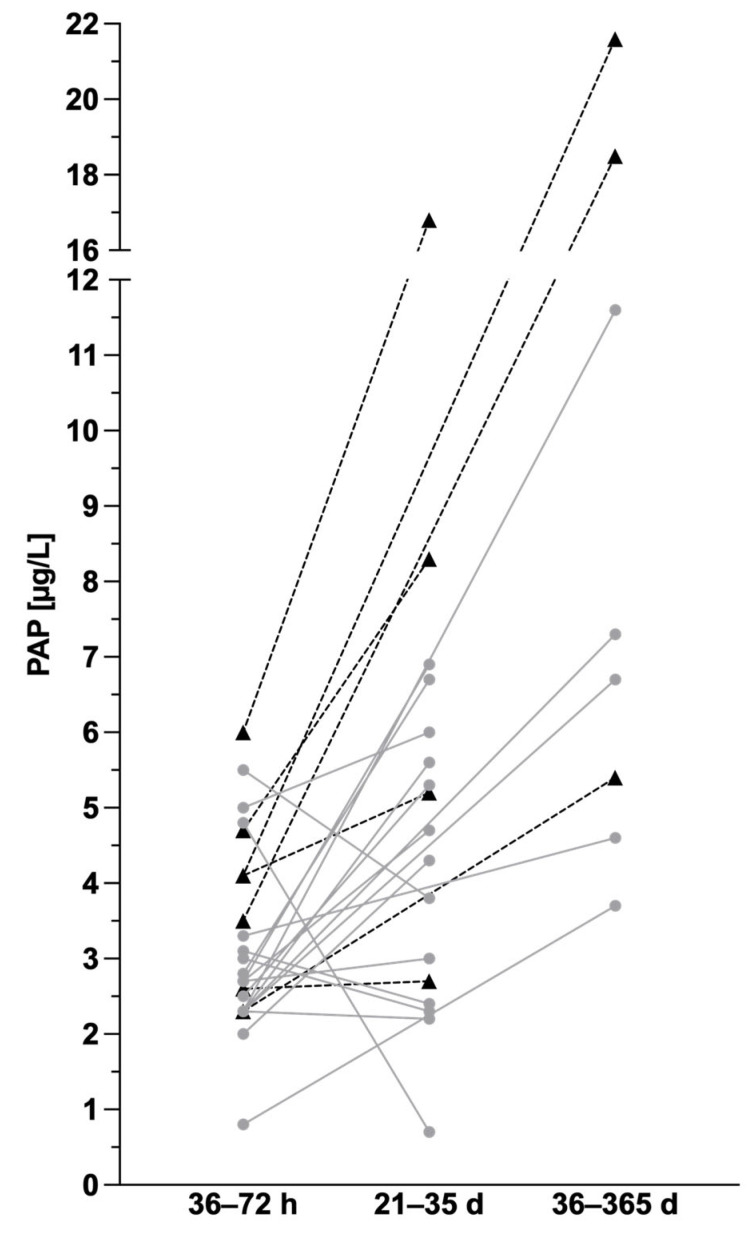
Longitudinal course of PAP values in healthy infants (grey lines, n = 17) and infants with cystic fibrosis (black dotted lines, n = 7) recruited for the Late-IRT&PAP study, of whom PAP values from CF-NBS were available.

**Table 1 IJNS-10-00005-t001:** Comparison of different percentiles of IRT concentrations in DBS in the CF-NBS in summer and winter 2009 to 2013. Summer includes the months June, July and August of the respective year “X”, winter includes the months December of the year “X” and January and February of the year “X + 1”.

Year	Percentile	IRT in Summer (µg/L)	IRT in Winter (µg/L)	*p*-Value
2009	95.0	42	46	<0.001
	99.0	63	70	<0.001
	99.9	131	146	0.584
2010	95.0	44	46	<0.001
	99.0	63	75	<0.001
	99.9	136	186	0.124
2011	95.0	43	44	<0.05
	99.0	61	67	<0.05
	99.9	131	124	0.752
2012	95.0	44	45	0.272
	99.0	67	67	0.840
	99.9	173	142	0.164
2013	95.0	44	44	0.924
	99.0	64	65	0.588
	99.9	141	149	0.560

## Data Availability

Publication or accessibility of patient-related data beyond what is represented above is not permitted due to local data protection regulations and ethics guidelines.

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
