# Peer review of "Influence of Season, Storage Temperature and Time of Sample Collection in Pancreatitis-Associated Protein-Based Algorithms for Newborn Screening for Cystic Fibrosis"

_2409-515X, 2024, doi:10.3390/ijns10010005_

Round 1

Reviewer 1 Report

Comments and Suggestions for Authors

Dear Authors, 

This is a novel, concise and interesting study, which presents findings relevant to NBS programmes, which employ a IRT/PAP/DNA algorithm for their CF NBS. I would be grateful if you could address the following comments that I have:

Line 73: some programmes accept screening samples for much longer periods of time, for example in The Netherlands samples are accepted until 6 months of age; for these programmes is this study also most valuable! 

Line 104/105: were IRT measurements AutoDELFIA and GSP comparable? 

line143-146: the samples are being sourced from different collection periods of the screening for the reasons given. Were there any other relevant changes within the NBS programme within the sampling time periods that may affect results, for example filter card type, way in which the sample is taken?

Line 187/188: were the outpatients otherwise healthy subject or were there medical reasons for their hospital visit? If these subject had other medical conditions, what can we say about the IRT and PAP concentrations in the 'healthy population' between 21-35 and 36-365 days of life? This relates also to the explanation lines 278 where exclusions for severe disease are described. Milder disease is not mentioned, but neither is confirmed that the non-CF inclusions were otherwise considered healthy.

Table 1/Figure 1: is it possible to indicate what the average temperature for the winter and summer periods was 2009-2015 (in the region most relevant to the samples)? Did this change significantly from year to year? The impact of humidity is not addressed; is there a reason for this?

Line 330-331: your findings have no impact on programmes that use a floating cut-off as your programme does, but can you reach the same conclusion for programmes that use a fixed cut-off? Could programmes using a fixed cut-off expect more referrals to second tier PAP in the winter months?

Line 349-351: since a decrease in PAP concentration is observed in samples older that 8 days at RT, for samples which are >8 days in transit to the screening laboratory, would you recommend applying different cut-off?  By not applying a different cut-off for samples >8 days do we risk reporting false negative results from screening?

Line 384-385: I think that the conclusion that PAP is not a sensitive marker if the sample is obtained <36 hours is too quickly made! This could be resolved by determination of PAP concentrations in the neonate population for each programme (as you state in line 390-391). I disagree with your assumption that this will make CF-NBS for individual programmes more complex (line 392), but I do agree that lowering PAP cut-off for programmes that sample <36 hours is only effective if the IRT/PAP/DNA algorithm is employed (line 397).

Table A3: I find the number of samples (5) for each condition investigated relatively small, but this is always a challenge in research related to rare disease. 

Author Response

Dear Reviewer,

Thank you for your excellent review, which has significantly improved the manuscript in many areas. We will respond to your comments point by point in the file uploaded!

Reviewer 2 Report

Comments and Suggestions for Authors

This manuscript is well written, and timely. The data will be a valuable addition to the literature.an important contribution to the literature. I have two recommendations.   I have two concerns and recommendation.   Please list the ambient temperature range and median for the area served by the Heidelberg newborn screening program during the June to August and December to February intervals. Did these change during the long duration of the study? For example, was there an overall warmer year in 2013 compared to 2009?Please specify how the dried blood by specimens are delivered to the newborn screening laboratory. Is this shipment accomplished by courier delivery so that the specimens are only exposed for less than 24 hours to ambient temperature variations? Or is the governmental postal service responsible for the deliveries, and, if so, what is the delay from collection of the specimens until they reach the newborn screening laboratory?

Author Response

Dear Reviewer,

Thank you for your excellent review. We will respond to your comments point by point within the file uploaded!
